# SARS-CoV-2 Viral Load, IFNλ Polymorphisms and the Course of COVID-19: An Observational Study

**DOI:** 10.3390/jcm9103315

**Published:** 2020-10-15

**Authors:** Emanuele Amodio, Rosaria Maria Pipitone, Stefania Grimaudo, Palmira Immordino, Carmelo Massimo Maida, Tullio Prestileo, Vincenzo Restivo, Fabio Tramuto, Francesco Vitale, Antonio Craxì, Alessandra Casuccio

**Affiliations:** 1Department of Health Promotion, Mother and Child Care, Internal Medicine and Medical Specialties, University of Palermo, 90127 Palermo, Italy; rosariamaria.pipitone@unipa.it (R.M.P.); stefania.grimaudo@unipa.it (S.G.); palmira.immordino@gmail.com (P.I.); carmelo.maida@unipa.it (C.M.M.); vincenzo.restivo@unipa.it (V.R.); fabio.tramuto@unipa.it (F.T.); francesco.vitale@unipa.it (F.V.); antonio.craxi@unipa.it (A.C.); alessandra.casuccio@unipa.it (A.C.); 2Department of Medicine, ARNAS Civico, Unità Operativa Malattie Infettive, 90127 Palermo, Italy; tullioprestileo@virgilio.it

**Keywords:** SARS-CoV-2, COVID-19 outcomes, viral load, single nucleotide polymorphisms, IFNλs

## Abstract

The course of SARS-CoV-2 infection ranges from asymptomatic to a multiorgan disease. In this observational study, we investigated SARS-CoV-2 infected subjects with defined outcomes, evaluating the relationship between viral load and single nucleotide polymorphisms of genes codifying for IFNλs (interferon). The study enrolled 381 patients with laboratory-confirmed SARS-CoV-2 infection. For each patient, a standardized form was filled including sociodemographic variables and clinical outcomes. The host’s gene polymorphisms (IFNL3 rs1297860 C/T and INFL4 rs368234815 TT/ΔG) and RtReal-Time PCR cycle threshold (PCR Ct) value on SARS-CoV-2 were assessed on nasal, pharyngeal or nasopharyngeal swabs. Higher viral loads were found in patients aged > 74 years and homozygous mutant polymorphisms DG in IFNL4 (adj-OR = 1.16, 95% CI = 1.01–1.34 and adj-OR = 1.24, 95% CI = 1.09–1.40, respectively). After adjusting for age and sex, a statistically significantly lower risk of hospitalization was observed in subjects with higher RtReal-Time PCR cycle threshold values (adj-OR = 0.95, 95% CI = 0.91, 0.99; *p* = 0.028). Our data support the correlation between SARS-CoV-2 load and disease severity, and suggest that IFNλ polymorphisms could affect the ability of the host to modulate viral infection without a clear impact on the outcome of COVID-19.

## 1. Introduction

As of 1 September 2020, globally, more than 26 million cases of coronavirus disease 2019 (COVID-19) have been reported from 215 countries, territories, areas and 2 international surveys [1]. In the WHO (World Health Organization) European Region, there have been more than 3 million cases reported, representing 21% of the global burden [2]. Although the pathology of COVID-19 is not yet fully understood, the infection may cause a wide spectrum of symptoms that varies from asymptomatic cases, mild symptoms of upper respiratory tract infections and life-threatening conditions. The severe acute respiratory syndrome coronavirus 2 (SARS-CoV-2) can affect respiratory, gastrointestinal and neurological systems, but there is growing evidence that other organs can also be impaired. According to a study from the China Center for Disease Control, 81% of people infected by SARS-CoV-2 had mild presentations while 14% had severe manifestations and about 5% had critical manifestations with needs for hospitalization and intensive care support [3]. Although the case fatality rate (CFR) has changed over time from the beginning of the outbreak and differs by location, an initial estimate suggests that CFR for SARS-CoV-2 is 2.58% [4]. However, it is difficult to calculate the CFR until the outbreak is over, and according to a systematic meta-analysis on the clinical features of COVID-19, in different settings the value may range from 3.75% to 13% [5].

Host factors could play a key role in determining clinical presentation and outcome of the infection. In fact, typical features of symptomatic patients infected by SARS-CoV-2 are disruption of the endothelial barrier, dysfunction of the alveolar capillary oxygen transmission and impairment of oxygen diffusion capacity [6].

A possible contribution of human gene polymorphisms involved in the host antiviral responses to SARS-CoV-2 has been recently postulated, and there is growing evidence that some gene polymorphisms could affect susceptibility and severity of the COVID-19 course [7]. Family clustering of severe cases has been reported worldwide, which supports further efforts in exploring these genetic determinants in order to be better prepared for future waves [8,9,10].

In particular, there is growing evidence that some type III or lambda interferons (IFNLs) signaling are involved in the regulation of immunity and in the antiviral sustained response [11]. The homozygosity for IFNλ3-IFNλ4 variants can be associated with a reduction of viral clearance in children affected by acute respiratory infections [12].

Another essential tool for managing and responding to the current COVID-19 pandemic is molecular testing. Clinical diagnosis of COVID-19 is confirmed by using the real-time reverse transcription polymerase chain reaction (rRT-PCR) to detect SARS-CoV-2 RNA [13]. The use of rRT-PCR as a molecular test allows to make qualitative (positive/negative results) diagnosis with, at least theoretically, a quantitative result that could be considered as a proxy of the viral load [14]. As a high viral load would require fewer rRT-PCR cycles, a low viral load would require many cycles to first record a significant fluorescent signal [15]. Data on the relationship between viral load and clinical outcome are still scarce, including viral load profiles at different times after diagnosis. A recently published review reports that the highest viral loads are detected at the time of symptom onset and generally decrease within one to three weeks after [16]. However, no clear evidence is available relating the infectivity to the presence of viral RNA detected with the rRT-PCR, as this does not indicate the presence of a live virus. Further findings have shown a correlation between cycle threshold levels and sample infectivity in a cell culture model [8]. In the present study, we investigated the relationship between cycle threshold values of quantitative rRT-PCR for SARS-CoV-2, presence of IFNL3/IFNL4 gene polymorphisms and the risk of severe outcomes (hospitalization, intensive care support or death) in COVID-19 patients.

## 2. Experimental Section

This is an observational study carried out between 24 February 2020 and 8 April 2020 on 383 consecutive patients whose respiratory biological sample swabs had been sent to the referral Laboratory for COVID-19 Surveillance for Western Sicily located at University Hospital “P. Giaccone” of Palermo. All patients included were laboratory-confirmed SARS-CoV-2 cases that had a positive result of a reverse transcriptase real-time polymerase chain reaction (rtReal-Time PCR) of nasal, pharyngeal or nasopharyngeal swabs according to the Centers for Disease Control and Prevention protocol [17]. For each patient, a standardized form was filled including sociodemographic variables (age, sex and residency), whereas clinical outcomes (home isolation, hospitalization, admission to intensive care unit and death) were obtained by consulting the clinical profiles centrally provided by the Italian National Institute of Health (Istituto Superiore di Sanità, ISS) and, when available, by direct contact with the hospitals involved in the care of each recruited patient. Outcome categories have been considered mutually exclusive and the worst outcome was reported for each patient.

Each patient was monitored for at least 21 days after recruitment, and the final day of follow-up was 8 April 2020. Due to the observational design (linked to referral), no follow-up biological samples were available. For two patients the rtReal-Time PCR cycle threshold value was not available and they were thus excluded.

Before the swab sampling, an individual informed consent was obtained from each patient by the health care provider. An approval to conduct the study was required and obtained from the Ethical Committee of the A.O.U.P. “P. Giaccone” of Palermo, Italy. The research reported in this paper is in accordance with the World Medical Association Declaration of Helsinki on Ethical Principles for Medical Research Involving Human Subjects.

### 2.1. SNP Genotyping

SNP (single nucleotide polymorphism) genotyping was carried out on the extracted whole nucleic acids from nasal or pharyngeal swabs (QIAamp Viral RNA Mini Kit, QIAGEN) by the TaqMan genotyping allelic discrimination method (StepOne Plus Real Time PCR System, A.B. Foster City, CA, USA) using custom genotyping assays (Thermo Fisher Scientific, Waltham, Massachusetts MA, USA). Complete genotyping was not possible for all patients due to the scarce amount of DNA available from swabs. The genotyping call was done by 2.3 Applied Biosystem Software (QIAGEN, Hilden, Germany). Genotyping was conducted in a blinded fashion relative to patient characteristics. Before testing for SNPs, samples were anonymized, and a unique randomly generated identification code was assigned to each record and to the correspondent swab. Researchers performing genetic analyses were unable to identify patients at all stages, and no permanent record linking these data to patient IDs was produced.

### 2.2. Statistical Analysis

The normality distribution of continuous variables was assessed by the Shapiro–Wilk test. Non-normal distributed continuous variables are presented as median and interquartile range (IQR), and categorical variables are expressed as the number of patients (percentage).

The Mann–Whitney rank sum test or the ANOVA test were used to compare non-parametric continuous variables between subgroups. Chi-square, Fisher exact tests and Fisher–Freeman–Halton tests were used for categorical variables as appropriate.

The multivariable logistic model was built to determine the association between RtReal-Time PCR cycle threshold values < 26 (the value of the I quartile) and homozygous mutant polymorphisms after adjustment for age. A multinomial regression model that was developed to test the relationship between clinical outcome (death/critical care or hospitalization vs. home isolation) and independent variables (sex, age and RtReal-Time PCR cycle threshold values) found statistically significantly associated to the outcomes at the univariate analysis. All statistical tests were two-tailed, and statistical significance was defined as *p* ≤ 0.05. Analyses were performed using R Software analysis 3.6.1 (Vienna, Austria) [18].

## 3. Results

The general characteristics of the study patients are summarized in Table 1. Overall, 381 patients with a median age of 58 years were evaluated. Death occurred in 32 (8.4%), whereas a large majority of subjects were isolated at home (235, 61.7%). The frequency of homozygous subjects for the variant polymorphism was 10.8% for TT in IFNL3 and 11.3% for DG in INFL4, being in high linkage disequilibrium.

In Table 2, median RtReal-Time PCR cycle threshold values were compared according to age, sex and the investigated polymorphisms.

A logistic regression analysis was computed on independent variables associated with RtReal-Time PCR cycle threshold values < 26. In this multivariable analysis, age > 74 years and IFNL4 DG homozygosity maintained their significance (adj-OR = 1.16, 95% CI = 1.01–1.34 and adj-OR = 1.24, 95% CI = 1.09–1.40, respectively).

In Table 3, the investigated variables are compared to clinical outcomes (death or intensive care support, hospitalization vs. home isolation).

The variables found statistically significantly associated with clinical outcomes were included in a multinomial regression analysis reported in Table 4.

In particular, after adjusting for age and sex, a significantly lower risk of hospitalization was found for subjects with higher RtReal-Time PCR cycle threshold values (adj-OR = 0.95, 95% CI = 0.91, 0.99, *p* = 0.028).

## 4. Discussion

A better understanding of the pathogenesis of SARS-Cov-2 induced direct and indirect damage in the host is needed in order to manage COVID-19. We explored two factors potentially related to diverse clinical outcomes, i.e., viral load estimated by a proxy (cycle threshold value of quantitative rRT-PCR for SARS-CoV-2 RNA) and single nucleotide polymorphisms (SNPs) in genes codifying for IFNλs (IFNL3 and IFNL4) that are major components of the innate immune response system.

The first intriguing finding of our study was that lower PCR Ct values and hence supposedly higher loads of SARS-Cov-2 were observed in older patients, in those with more severe outcomes and in the presence of DG homozygosity for INFL4. These associations should be considered with caution, since they have been found in two different multivariable models and, thus, a direct linkage between the three variables can be only inferred. However, a more in-depth reasoning could help to better clarify our hypotheses and these relations.

In regard to age, our study confirms previous, well-substantiated evidence that older age is linked to a worse outcome [19,20]. It must be stressed that we did not assess the role of pre-existing comorbid conditions such as hypertension, cardiovascular disease and diabetes due to lack of data in many of these patients, hence it cannot be excluded that old age as a risk factor also acts as a proxy of such comorbidities. Older age in our cohort was significantly associated with a higher load of SARS-CoV-2, an association recently reported by other authors [21,22]. A theoretical explanation for this association could be related to immunosenescence, which could impair innate and adaptive immune responses as age increases.

A low PCR Ct value in our group of patients was also correlated to a higher risk of admission to the hospital or to an intensive care unit and therefore of death. A correlation with death has been already observed by Chu et al. in relation with SARS-CoV-2 viral load [23]. In our analysis, after adjustment for potential confounding due to age and sex, only hospitalization maintained its significant correlation to viral load. It must be stressed that our estimate by proxy of SARS-Cov-2 load relates to the first detection of SARS-CoV-2 in patients, whereas death, in almost all deceased patients, occurred days or even weeks after first detection of the virus. This makes a direct causal relationship between the two variables quite unlikely. Conversely, after adjusting for age and sex, a 5% reduction of the risk of hospitalization was observed for each unit increase of PCR Ct values. Other authors have suggested a relationship between viral load, measured as PCR Ct, and clinical severity of disease [24], although these findings require further investigation.

As is already well documented [25,26,27], we found a close association between male sex and severity of COVID-19, both as hospitalization and death/intensive care. Some authors have hypothesized the role of potential sex-specific mechanisms modulating the course of the disease, such as hormone-regulated expression of some genes, as well as sex hormone-driven innate and adaptive immune responses and immune aging [25]. However, other concurring risk factors including comorbidities, gender-specific lifestyle or health behaviour could interact and increase this risk. Being a novel virus, little is known on the impact of host genetics on SARS-CoV-2 infection and clinical presentation. While much research has focused on viral receptors and several genetic associations between ACE2 genetic variants and the increased patient susceptibility to the infection, limited data exist regarding all the other genes that have been implicated in the pathology of the disease. It has been shown that the infection causes lymphopenia due to impairment of lymphopoiesis and increases T lymphocyte apoptosis, with a strong inflammatory response including a massive release of cytokines precipitating the alveolar damage and causing multiorgan failure [28]. In the majority of severe cases, the increase of serum concentrations of cytokines, including IL-2R, IL-6, TNF-α and IL-l, leads to a “cytokine storm” probably associated with severe cases and high mortality [28]. As mentioned earlier, considering the underlying genomic potential susceptibility to the infection, we have assessed host gene polymorphisms by using nucleic acid extracts generated from swabs during the diagnostic processing of COVID-19 to evaluate the host’s genetic profile. Although no clear pathogenetic pathway can be defined, on the basis of our data, we have assessed that DG homozygous polymorphic status in INFL4 is significantly associated with a 16% increased risk of higher viral load (measured as PCR Ct value < 26).

This evidence is in keeping with the key role of IFNλs in the response to viral infection sustained by negative- and positive-sense RNA virus, double stranded RNA viruses and DNA viruses. As proof of the key role of type III IFNs in the regulation of immunity response, single nucleotide polymorphisms in IFNλ genes were strongly associated with outcomes of viral infection [10]. Between them, the rs1297860(C/T) located ~3 kb upstream of IFNL3 and the frame shift variant rs368234815 (TT/ΔG), which are in high linkage disequilibrium, represent the strongest host factor associated with viral clearance. It has been reported that the homozygosity for IFNλ3-IFNλ4 variants, overrepresented in African descent, is associated with a reduction of viral clearance in children affected by acute respiratory infections, including Rhinovirus and Coronavirus [11]. Very recently, it has also been reported that a subset of patients with life-threatening COVID-19 pneumonia were characterized by neutralizing auto Abs against type I IFNs, as well as patients with inborn errors of type I IFNs [29,30]. This evidence could suggest the protective role of type I IFNs signaling against severe SARS-CoV-2 disease.

According to this evidence, the use of IFNλs as an antiviral drug has been suggested in COVID-19 patients or in subjects at high infection risk, and current randomized clinical trials with peg-IFN L1 are ongoing [31].

It is conceivable that subjects with rs1297860 TT and rs368234815 DG/DG haplotype show a lower ability in virus clearance (low PCR Ct values), associated with the defective upregulation of inflammatory pathways (low risk for hospitalization due mainly to inflammation damage).

All of the previous findings should be considered with caution, since this study has several potential limitations. In particular, our sample size is relatively small to evaluate with appropriate strength, complex associations and interactions. However, a major limit of small sample sizes is the chance of type II error, thus, since we could have lost some associations, our results can be interpreted as exploratory and descriptive. Moreover, we used PCR Ct values, such as proxy of viral load, although this relationship is still not well standardized and quantified and, thus, could be affected by a low degree of precision and accuracy. Finally, we did not check for other potential confounding factors (such as comorbidities) that could be influencing the results. We also did not exclude the possibility that some hospitalizations were due to infection control reasons. For this reason, we have considered hospitalized subjects (those at higher misclassification risk) with caution by performing a multinomial analysis where home isolation was the reference. In this way, if results obtained for hospitalized subjects should also be biased, both home isolation and death/critical care would be poorly influenced by this bias.

## 5. Conclusions

Despite possible limitations, we are confident that our findings contribute to the search for some pieces of this very complex puzzle and suggest an interesting link between SARS-CoV-2 load, estimated by RtReal-Time PCR cycle threshold value at presentation, severity of COVID-19 and specific IFNλ polymorphisms affecting the ability of the host to modulate viral infection in the early stages, thus acting as regulators of the ultimate outcome of COVID-19.

## Figures and Tables

**Table 1 jcm-09-03315-t001:** General characteristics of the patients.

		All Patients (*n* = 381)	
**Male sex, *n* (%)**	Male	205	(53.8%)
**Age in years, median (IQR)**		58	(44–74)
**RtReal-Time PCR cycle threshold, median (IQR)**		31	(26–35)
**Main clinical outcome, *n* (%)**			
	Death	32	(8.4%)
	Intensive care hospitalization	21	(5.5%)
	Hospitalization	93	(24.4%)
	Home isolation	235	(61.7%)
**IFNL3 rs12979860 C > T *n* (%)**			
	CC	158	(41.5%)
	TC	182	(47.8%)
	TT	41	(10.8%)
**INFL4 rs368234815 TT/DG *, *n* (%)**			
	DG/DG	34	(11.3%)
	TT/DG	144	(48.0%)
	TT/TT	122	(40.7%)

IQR: interquartile range; * Not assessed in all subjects.

**Table 2 jcm-09-03315-t002:** Univariate analyses on RtReal-Time PCR cycle threshold values and investigated independent variables.

		RtReal-Time PCR Cycle Threshold Values, Median (IQR)		*p*-Value
**Sex**				
	Male	31	(26–36)	0.39
	Female	31	(25–35)	
**Age in years**				
	0 to 29	32	(29–37)	<0.001
	30 to 44	31	(26–35)	
	45 to 64	32	(29–36)	
	65 to 74	30	(26–35)	
	>74	28	(23–31)	
**IFNL3 rs12979860 C > T**				
	CC	31	(25–36)	0.08
	TC	31	(27–35)	
	TT	29	(22–34)	
**INFL4 rs368234815 TT/DG ***				
	DG/DG	29	(24–32)	0.03
	TT/DG	31	(27–35)	
	TT/TT	30	(25–35)	

IQR: interquartile range; * Not assessed in all subjects.

**Table 3 jcm-09-03315-t003:** Univariate analyses on predictors of clinical outcomes.

		Death/Critical Care	Hospitalization	Home Isolation	*p*-Value
**Sex, *n* (%)**					<0.001
	Male	37 (18.0%)	57 (27.8%)	111 (54.1%)	
	Female	16 (9.9%)	39 (22.2%)	121 (68.8%)	
**Age in years, median (IQR)**		75 (67–79)	63 (53–76)	51 (39–66)	<0.001
**RtReal-Time PCR cycle threshold, median (IQR)**		29 (25–32)	29 (25–34)	32 (27–36)	<0.001
**IFNL3 rs12979860 C > T, *n* (%)**					
	CC	24 (15.2)	33 (20.9)	101 (64.0)	0.46
	TC	24 (13.2)	49 (27.0)	109 (59.9)	
	TT	5 (12.2)	14 (34.2)	22 (53.7)	
**INFL4 rs368234815 TT/DG *, *n* (%)**					
	DG/DG	4 (11.8)	13 (38.2)	17 (50.0)	0.30
	TT/DG	21 (14.6)	37 (25.7)	86 (59.7)	
	TT/TT	22 (18.0)	25 (20.5)	75 (61.5)	

IQR: interquartile range; * Not assessed in all subjects.

**Table 4 jcm-09-03315-t004:** Multinomial regression analyses on variables associated with major clinical outcomes (death/critical care or hospitalization vs. home isolation).

	Death/Critical Care OR (95% CI)	Hospitalization OR (95% CI)
**Sex (ref. Female)**	4.11 (1.98,8.53) ***	1.99 (1.18,3.35) **
**Age in years**	1.07 (1.05,1.1) ***	1.03 (1.01,1.04) ***
**RtReal-Time PCR cycle threshold value**	0.66 (0.33,1.32)	0.95 (0.91,0.99) *

* < 0.05; ** < 0.01; *** < 0.001.

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
