# Peer review of "SARS-CoV-2 Viral Load, IFNλ Polymorphisms and the Course of COVID-19: An Observational Study"

_jcm, 2020, doi:10.3390/jcm9103315_

Round 1

Reviewer 1 Report

This study assesses associations between CT values for SARS-CoV 2, age, gender and clinical outcome. While it identifies an association between age and CT value, as well as INFL4 rs368234815 TT/DG* and CT value, INFL4 rs368234815 TT/DG* is not associated with a worse clinical outcome. The article is very well written and requires only minor adaptations.

Introduction: From the introduction, it is not clear, why  IFNL3/IFNL4 gene polymorphisms were chosen as targets. The immunological context should be explained.

Ethics: Were patients explicitly informed and consented to analysis of their genes? You may want to add a short note on this.

In the regression model you compared Death/Critical care or Hospitalization vs. Home isolation. Sometimes, patients are hospitalized for infection control reasons, e.g. when they have persons at risk in their family at home. Would these patients not distort the analysis? How did you deal with this aspect?

In the results section, a very succinct summary highlighting the relevant findings in the tables without repeating them would be helpful.

Table 1) Are the categories Death, Intensive care hospitalization, hospitalization and home isolation overlapping? If yes, please state so in the text or table

All Tables: It is not necessary to list male and female data, as they are complementary.

While you state that INFL4 rs368234815 TT/DG* was associated with a low CT value, it was not associated with a worse clinical outcome. Therefore, the clinical value of this finding is rather questionable. This should be stressed further in the abstract and in the discussion.

Reviewer 2 Report

Initially, on reading the title and abstract of this paper, I was enthusiastic for its findings given recent reports regarding COVID-19 and auto-antibodies against type I IFNs (Bastard P et al, PMID: 32972996) and inborn errors of type I IFN (Zhang Q et al, PMID: 32972995). However, the methods and findings leave something to be desired and rather temper my initial enthusiasm. Within the limits of the observational study that the authors have set out to perform, they have provocative findings: that a surrogate measure of SARS-CoV-2 viral load is correlated to risk of hospitalization and may be correlated with IFN lamda polymorphisms. However, I do not see much novel information in this study compared to what is already published and because the surrogate marker for SARS-CoV-2 that was used (cycle count from nasal swabs) is fraught with control issues, I think the strength of the findings are significantly hampered.

More specific comments are:

  • The introduction would benefit from a more thorough discussion of the polymorphisms of IFNs and implications of finding certain polymorphisms overrepresented in patients in the study. In particular, the first paragraph of the results section introduces “TT” and “DG” with no preceding explanation of these polymorphism.
  • The papers mentioned above were released after this manuscript was submitted, so it is understandable that the authors would not be able to reference them, but it would be worth acknowledging their findings and comparing the results of those studies to the authors’ findings.
  • Line 38-39, when introducing the PCR cycle thresholds, the source material (i.e. nasopharyngeal swabs, etc) should be stated

I am not a fan of blanket statements that the manuscript needs an English language editor. In fact, I find the English very readable and cogent. I do think the following edits would help readability, however:

Line 34: “relation” should be “relationship”

Line 35: “codifying” should be “coding”

Line 37: “Host gene polymorphisms…” is probably preferred over “Host’s”

Line 55: “that” in lieu of “which”

Line 70: “contribute” should be “contribution”

Line 75: the second “the” is unnecessary, as is “diagnosis”

Line 78: insert “a” between “as” and “molecular”

Line 80: “less” should be “fewer”

Line 81: Change to “…first record a significant fluorescent signal.”

Line 111: period, not a slash at end of sentence.

Line 117: Insert “of” between “amount” and “DNA”

Line 150: Unclear what the symbol is between “age” and “74”

Line 226: Is there a word or several words missing from the end of this sentence?

Line 229: “haplotype” in stead of “aplotype”

Line 223: change “cautions” to “caution”

Line 240: insert “such” between before “as”
